# Audio ALBERT:
# A Lite BERT for Self-supervised Learning of Audio Representation

Po-Han Chi [1]  Pei-Hung Chung [1]  Tsung-Han Wu [1]  Chun-Cheng Hsieh [1]  Shang-Wen Li [2 3]  Hung-yi Lee [1]

## Abstract

For self-supervised speech processing, it is crucial to use pretrained models as speech representation extractors. In recent works, increasing the size of the model has been utilized in acoustic model training in order to achieve better performance. In this paper, we propose Audio ALBERT, a lite version of the self-supervised speech representation model. We use the representations with two downstream tasks, speaker identification, and phoneme classification. We show that Audio ALBERT is capable of achieving competitive performance with those huge models in the downstream tasks while utilizing 91% fewer parameters. Moreover, we use some simple *probing models* to measure how much the information of the speaker and phoneme is encoded in latent representations. In probing experiments, we find that the latent representations encode richer information of both phoneme and speaker than that of the last layer.

## 1. Introduction

Recently, pretrained models (Devlin et al., 2018; Peters et al., 2018; Radford et al., 2018; 2019), especially BERT, dominate Natural Language Processing (NLP) world. The models learn powerful and universal representation by utilizing self-supervised learning at pretraining stage to encode the contextual information. The representation is beneficial to performance, especially when the data of the downstream task is limited. As of late, BERT-like models are also applied to the speech processing domain. The pretraining

model learns the robust speech representations for speech processing tasks, for example, ASR and speaker recognition, with the self-supervised learning approaches (Liu et al., 2019a; Jiang et al., 2019; Ling et al., 2019; Baskar et al., 2019; Schneider et al., 2019). However, since the size of the pretraining models, no matter the text or speech versions is usually prohibitively large, they require a significant amount of memory for computation, even at the fine-tuning stage. The requirement hinders the application of pretrained models from different downstream tasks.

ALBERT (Lan et al., 2019) is a lite version of BERT for text by sharing one layer parameters across all layers and factorizing the embedding matrix to reduce most parameters. Although the number of parameters is reduced, the representations learned in ALBERT are still robust and task agnostic, such that ALBERT can achieve similar performance in the same downstream tasks comparing to BERT. In this paper, we bring the idea of sharing parameters from ALBERT to the speech processing domain and propose a novel self-supervised model, Audio ALBERT (AALBERT). AALBERT shows comparable performance to other pretrained models on downstream tasks, but with much smaller models.

Besides showing performance, we further analyze representations extracted from different layers of the model. We use a simple classifier to *probe* each layer, and we find that the representations of the intermediate layers contain more phonetic and speaker information than that of the last layer, which indicates that the representations extracted from the last layer fit the pretraining task too much. As a result, they may be unsuitable for downstream tasks comparing to those from the intermediate layers.

## 2. Related work

### 2.1. Self-supervised learning representation

In recent years, some works related to self-supervised learning spring up in Computer Vision (CV), NLP, speech processing, etc. In CV, some works (Chen et al., 2020; He et al., 2019) incorporate contrastive objective and self-supervised learning for learning visual representation. In NLP, some works also utilize self-supervised learning to learn language

[1] College of Electrical Engineering and Computer Science, National Taiwan University [2] Amazon AI [3] work done before joining Amazon. Correspondence to: Po-Han Chi <r08942074@ntu.edu.tw>, Pei-Hung Chung <r05942048@g.ntu.edu.tw>, Tsung-Han Wu <r07942145@ntu.edu.tw>, Chun-Cheng Hsieh <r07942150@ntu.edu.tw>, Shang-Wen Li <shangwei@amazon.com>, Hung-yi Lee <hungyilee@ntu.edu.tw>.

*Proceedings of the 37th International Conference on Machine Learning*, Vienna, Austria, PMLR 119, 2020. Copyright 2020 by the author(s).

representations. ELMo (Peters et al., 2018) is the first work introducing the concept of contextualized embeddings and the weighted sum application. BERT (Devlin et al., 2018) is the first work introducing the concept of masked language model with deep transformer encoder architecture. Masked Language Model (MLM) is one of the novelties proposed in BERT; it has to reconstruct the masked input sequences in the pretraining stage. XLNet(Yang et al., 2019), built with different attention mechanisms, outperforms than both autoregressive models and MLM.

However, Roberta (Liu et al., 2019b), a BERT model with more data, larger batch size, and the better hyperparameters, shows the competitive results with XLNET in different downstream tasks. Last but not least, ALBERT (Lan et al., 2019) reduces the parameters drastically without losing performances on downstream tasks comparing to BERT.

### 2.2. Speech representation

Contrastive Predictive Coding (CPC) (Oord et al., 2018) incorporates contrastive objective in self-supervised learning to learn powerful representations in many fields. Autoregressive Predictive Coding (APC) (Chung et al., 2019) leverages the idea of an autoregressive model from ELMo to learn stronger speech representations. Inspired by MLM, Mockingjay (Liu et al., 2019a) masks frame in input acoustic feature and tries to reconstruct the corresponding linear spectrogram or mel spectrogram in the pretraining stage. Similarly, Masked Predictive Coding (MPC) (Jiang et al., 2019) uses the idea of MLM to pretrain a model for speech recognition. Speech-XLNet (Song et al., 2019) is the audio version of XLNet. vq-wav2vec (Baevski et al., 2019) incorporates vector quantization and BERT to improve the performance on downstream tasks.

Finally, DeCoAR (Ling et al., 2019), a pretrained LSTM model, performs well in applying the representation on speech recognition task which build from deep LSTM module also use a similar task like Mockingjay and MPC in the pretraining stage. To sum up, All pretrained model size is large in common, which motivates us to build a lite version of pretrained model.

### 2.3. Probing task

Probing is a technique to measure whether the encoder embeds specific information in representation (Jawahar et al., 2019; Belinkov et al., 2019; Li et al., 2020). The probing can be done by extracting representation we want to examine, applying it in a downstream probing model, and measuring the performance. A method is proposed to synthesize audio from the ASR hidden state (Li et al., 2020), which can be considered as another way of probing.

*Table 1.* Pretrained Models

| MODEL | LAYER | PARAM | PARAM SHARING |
|---|---|---|---|
| AALBERT-12L | 12 | 7.4M | TRUE |
| AALBERT-6L | 6 | 7.4M | TRUE |
| AALBERT-3L | 3 | 7.4M | TRUE |
| MOCKINGJAY-12L | 12 | 85.4M | FALSE |
| MOCKINGJAY-6L | 6 | 42.8M | FALSE |
| MOCKINGJAY-3L | 3 | 21.4M | FALSE |

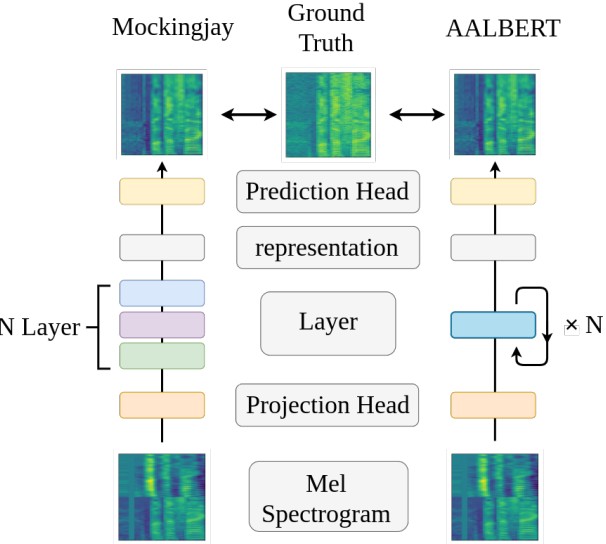

*Figure 1.* Difference between Mockingjay and AALBERT

## 3. AALBERT

### 3.1. Pretraining

At the pretraining stage, we feed the AALBERT with 160-dimension hidden state. Each hidden states contains 80-dimension mel spectrogram along with its delta as the input, and train the networks to reconstruct the corresponding linear spectrogram from the masked input. For simplicity, we denote the input as input acoustic feature after this section. We apply the masking to each utterance by first downsampling one out of every three frames and then randomly selecting 15% of the resulting frames for masking. We mask selected frames to zero with 80% probability, replace with other random frames from the utterance with 10% probability, and keep the original frames for the remaining cases.

Figure 1 shows the difference between AALBERT and other models pretrained on audio like Mockingjay (Liu et al., 2019a). The main difference is that AALBERT shares the same parameters across layers, resulting in having much fewer network parameters.

In the pretraining stage, we train our model with learning

rate 5e-5, batch size 50, and Lamb optimizer (You et al., 2019) for approximate 500k steps. The models are trained on a single NVIDIA Tesla V100 32GB. In Table 1, we show the information of all pretrained models used in this paper.

## 3.2. Downstream tasks

There are a variety of ways to apply a pretrained model to downstream tasks. They can be categorized into two ways.

### 3.2.1. FEATURE EXTRACTION

In feature extraction, all parameters in the pretrained models are fixed when training on the downstream tasks. Here we utilize the representations extracted from the pretrained model as fixed features and feed them into a simple, trainable classifier. A typical implementation is to use the representations of the last layer as features. On the other hand, there is yet another weighted sum approach, which is proposed by ELMo (Peters et al., 2018). To train the models on the downstream tasks, we use all the representations extracted from the different layers rather than the last one only. Note that the weights here are some learnable parameters.

### 3.2.2. FINE-TUNING

For fine-tuning, the whole model, including both AALBERT and those layers for downstream tasks, is trainable. This technique can boost the model performance dramatically on difficult tasks such as phoneme classification. For the simple tasks, the setup in Section 3.2.1 is adequate.

*Table 2.* Hyperparameter for different downstream tasks, BS: Batch size, LR: Learning rate

| DOWNSTREAM | DETAILS | LR | BS |
|---|---|---|---|
| PHONEME CLASSIFICATION | WEIGHTED-SUM | 1E-3 | 48 |
| | FINE-TUNED | 1E-4 | 12 |
| UTTERANCE-LEVEL SPEAKER IDENTIFICATION | 921 SPEAKERS | 1E-3 | 48 |
| | 251 SPEAKERS | 1E-3 | 48 |
| FRAME-LEVEL SPEAKER IDENTIFICATION | 251 SPEAKERS | 1E-3 | 48 |

## 4. Experiments

We evaluate the quality of those different features extracted from our pretrained AALBERT on several downstream tasks, including one phoneme classification task and three speaker identification tasks. For different downstream tasks, we apply different downstream models trained with different hyperparameters. The detailed hyperparameters for each downstream tasks are in Table 2, and the model architecture of the downstream models would be elaborated in the following subsections.

### 4.1. Phoneme classification

To measure the phonetic information, we train 2-layer phoneme classifiers, whose input takes the representations generated from Mockingjay (Liu et al., 2019a) or AALBERT, both trained on the train-clean-360 subset of LibriSpeech (Panayotov et al., 2015). Then, we obtain the force-aligned phoneme sequences, which contains 72 possible phone classes, with Montreal Forced Aligner (McAuliffe et al., 2017).

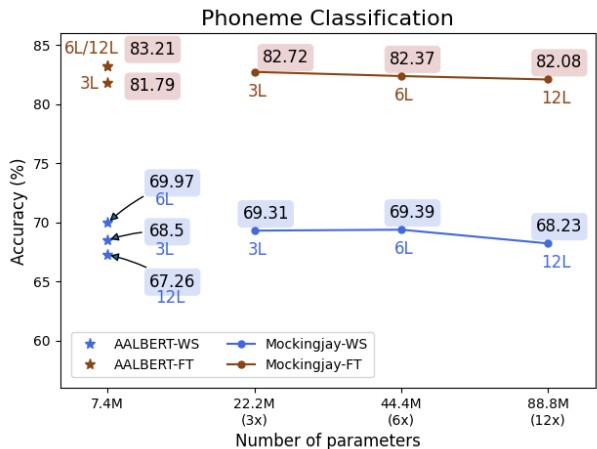

*Figure 2.* Phoneme accuracy on different models with their model parameters. "WS":settings utilizing the weighted-sum representation; "FT": settings of fine-tune stage.

In Figure 2, we show the performance of our models with different layers and settings and compare them to the baseline model (Mockingjay). The vertical axis is the phoneme classification accuracy, while the horizontal axis is the number of network parameters. For both fine-tuning case and weighted-sum case, our models show the classification accuracy compared to Mockingjay, but with much fewer network parameters. Also, note that AALBERT-12L does not perform well; this might be due partially to the limited data and the sharing-parameter mechanism in ALBERT. AALBERT-12L is too deep to optimize by a limited amount of data, not to speak of sharing parameters across layers. In this situation, the shallower model, AALBERT-3L and AALBERT-6L, would be adequate.

In Figure 3a and Figure 3b, we show the performance on phoneme classification tasks of both feature-extraction case and fine-tuning case versus different proportions of training data being used. Here are two observations. First of all, not only Mockingjay but AALBERT outperforms the input acoustic feature (shown in Figure 3a, Figure 3b). Secondly, these figures show that the representations extracted from Mockingjay and AALBERT have similar performance on phoneme classification tasks.

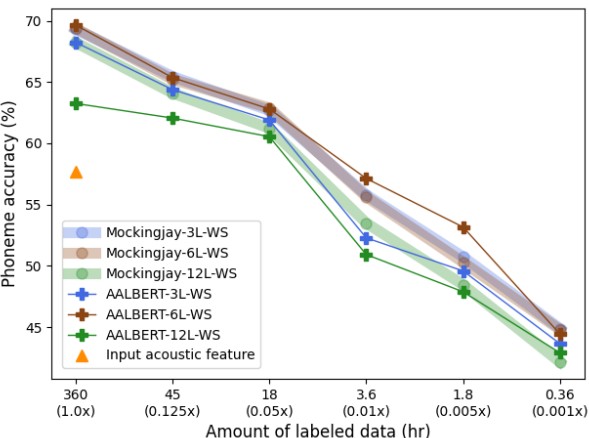

(a) Feature-extraction case

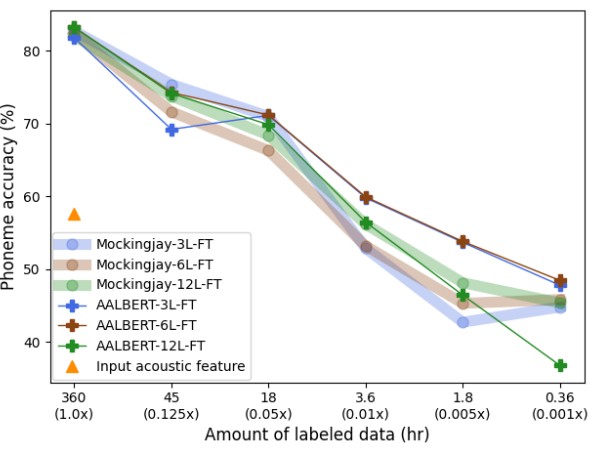

(b) Fine-tuning case

Figure 3. Phoneme classification accuracy vs amount of labeled data. 3L, 6L, 12L: the number of layers, FT: fine-tune, WS: weighted sum, Input acoustic feature: input acoustic feature as baseline.

## 4.2. Speaker identification

We evaluate the model performance with two tasks, utterance-level and frame-level here.

1. Utterance-level speaker identification: Classifying speakers in **train-clean-100** and **train-clean-360**

2. Frame-level speaker identification: Classifying speakers in **train-clean-100** only.

There are 921 speakers in the Librispeech train-clean-360 subset and 251 speakers in the Librispeech train-clean-100 subset. We only use the weighted-sum representations in this part due to space limitation. Besides, in the previous work (Liu et al., 2019a), the speakers with few training data

are filtered out in the experiments, yet in this paper, we use all data in these two LibriSpeech subsets.

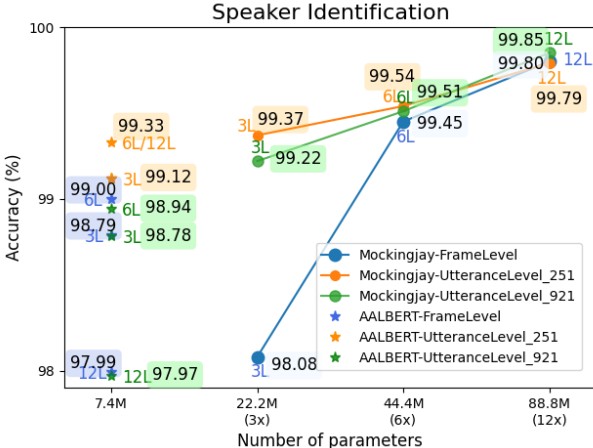

Figure 4. Speaker accuracy on different models and settings with their model parameters. "FrameLevel": settings of frame-level speaker identification, "UtteranceLevel_251": settings of utterance-level speaker identification on 251 speaker, "UtteranceLevel_921": settings of utterance-level speaker identification on 921 speaker.

### 4.2.1. UTTERANCE-LEVEL SPEAKER IDENTIFICATION

We split both datasets into training, development, and test set in the ratio of 8:1:1. Besides, the model here is a linear classifier with a mean-pooling layer. In Figure 4, it shows that AALBERT is competitive against Mockingjay, both of which are much better than the baseline (input acoustic feature, the accuracy is about $0.59\%$ here). The results show that both AALBERT and Mockingjay encode much richer speaker information than the baseline method.

Furthermore, we use t-SNE (Maaten & Hinton, 2008) to visualize the utterance representations extracted from input acoustic feature and AALBERT in Figure 5a and Figure 5b. In the figures, each point represents an utterance, which is generated by the mean-pooling layer; different speakers have different colors here. The representations from AALBERT are clustered together, and the elements in the same cluster represent exactly the same speaker. On the other hand, we cannot observe the same phenomenon on the input acoustic features, which shows that AALBERT may encode much speaker information.

### 4.2.2. FRAME-LEVEL SPEAKER IDENTIFICATION

For a fair comparison with Contrastive Predictive Code (CPC) (Oord et al., 2018), we split the data in the same way with it and only report the results in the frame-level setting. The model here is a simple linear classifier. Figure 4 shows that not only AALBERT but Mockingjay outperforms CPC $(97.04\%)$ and the input acoustic features $(0.3\%)$.

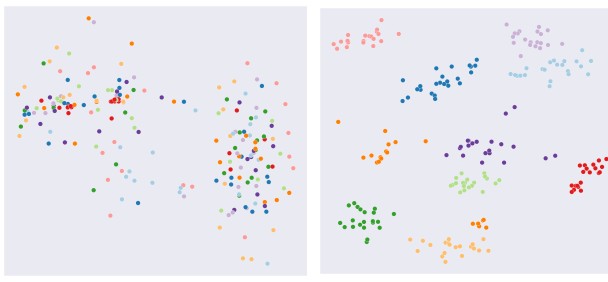

(a) Input acoustic feature          (b) AALBERT representations

*Figure 5.* Visualization of 10 speakers representations via t-SNE. Different colors represent different speakers.

In conclusion, AALBERT shows comparable results on speaker identification tasks against Mockingjay, yet using 91% fewer parameters.

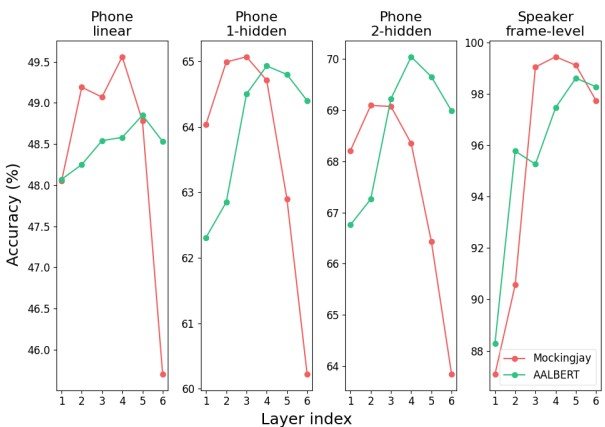

*Figure 6.* Probing task of AALBERT-6L and Mockingjay-6L

## 4.3. Probing task

We utilize two probing tasks, phoneme classification and frame-level speaker identification[1], to examine how much phoneme and speaker information contain in the representations of each layer. In both tasks, we use train-clean-100 dataset, which is unseen at the pretraining stage. We probe the AALBERT-6L and Mockingjay-6L since the average performances of them are the best. We utilize three different classifiers as the probing models, linear, one-hidden layer, and two-hidden layer, to probe each layer of the pretrained models for the speaker information and the phoneme information. We used several probing models with different network architectures to mitigate the possible bias from the probing models.

---

[1]Since we want to analyze an individual representation instead of the whole utterance, we choose frame-level instead of utterance-level.

Figure 6 shows the result of probing tasks. For the probing of phoneme information, the three different probing models show the same trends among the same pretraining model. In both pretraining models, as the depth increases, the phoneme information increases first and then decreases. Comparing the two pretraining models, the peak of the Mockingjay-6L is closer to the input layers than AALBERT-6L. On the other hand, when comparing the absolute performance of Mockingjay-6L and AALBERT-6L, the concluding from different probing models would be different. Mockingjay-6L achieves better phoneme classification accuracy for the shallower probing model, whereas AALBERT-6L obtains better performance of the deeper probing model. For speaker information, the $5^{th}$ layer of AALBERT-6L contains the most speaker information, while the $4^{th}$ layer is the best for Mockingjay-6L.

The results in Figure 6 further indicate that the intermediate representations outperform the representations from the last layer in all four different probing tasks regardless of Mockingjay-6L or AALBERT-6L model. This might indicate that the last layer fits the pretraining tasks too much; therefore, the representations extracted from the intermediate layers may be more suitable for downstream tasks.

## 5. Conclusion

In this paper, we present a novel model, Audio ALBERT (AALBERT). AALBERT is a pretrained model for extracting latent representations that encode the audio information. The model is learned by reconstructing the masked input acoustic features to the linear spectrogram. We show that AALBERT can achieve comparable performances against Mockingjay, a BERT-like pretrained audio model, yet with much fewer parameters. Besides, we show the promising results in encoding audio information with much smaller pretrained models. For our future work, we will investigate various model architectures to further improve the efficiency of pretrained models in computation and parameter usage.

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
