# OpenReview forum: "Audio ALBERT: A Lite BERT for Self-supervised Learning of Audio Representation"
_ICML.cc/2020/Workshop/SAS — Submitted to SAS 2020_

### Official Review · AnonReviewer2 · 2020-06-23
**Audio version of Bert that is more of an Audio Encoder**

**Rating:** 4
**Confidence:** 4

**Review:**

The model is essentially a Denoising Auto-Encoder using log-mel features. Inputs are zero-ed
frames, similar to frame-dropping or masked LM for NLP or CutOut for Vision Tasks.

The authors seem to have missed a long history in unsupervised training for
speech processing. Self-training for ASR was done 30 years ago, similar for
various forms of auto-encoders.

In figure 3.a, are the acoustic features 80logmel + deltas ? Does it include
Mean and Variance Normalization ? Please clarify.

The experimental setup with phoneme classification injects external knowledge
in form of the Forced Alignment model and makes it therefore difficult to judge your results.

Audio data is by nature of sequential nature. The way you set up the experiments,
you cannot show that your model learns any sequential structure.

Why don't you follow experimental setups that researchers use to test
un-supervised training approaches for ASR ? Remove the external forced alignment
model and build ASR models and see if your features have any impact.The model is a Denoising Auto-Encoder using log-mel features. Inputs are zero-ed
frames, similar to frame-dropping or masked LM for NLP or CutOut for Vision Tasks.

The authors seem to have missed a long history in unsupervised training for
speech processing. Self-training for ASR was done 30 years ago, similar for
various forms of auto-encoders.

In figure 3.a, are the acoustic features 80logmel + deltas ? Does it include
Mean and Variance Normalization ? Please clarify.

The experimental setup with phoneme classification injects external knowledge
in form of the Forced Alignment model and makes it therefore difficult to judge your results.

Audio data is by nature of sequential nature. The way you set up the experiments,
you cannot show that your model learns any sequential structure.

Why don't you follow experimental setups that researchers use to test
un-supervised training approaches for ASR ? Remove the external forced alignment
model and build ASR models and see if your features have any impact.

My main problem with the paper is that I don't see much novelty. BERT for text problems was novel, but this "Audio BERT" is
really just a denoising AudioEncoder and plenty of related work exists. Also, this type of features never got strong improvement over
classic unsupervised training for ASR. Also, the evaluation framework uses an external forced alignment tools that really does
most of the work, e.g. segmenting the audio.

---

### Official Review · AnonReviewer1 · 2020-06-26

**Rating:** 4
**Confidence:** 3

**Review:**

Authors applied previously introduced BERT-like architectures as well as a similar pre-training strategy to BERT in order to obtain representations of audio for downstream tasks in the standard application setting of self-supervision.

While authors report important empirical findings in that parameter-efficient models are shown to perform closely to much larger counterparts, my main concern with this contribution lies in the fact that there’s limited novelty other than the application in audio tasks of previously introduced approaches from other domains.

Some suggestions/questions are listed below regarding the empirical evaluation:

Baselines: while the comparisons between AALBERT and Mockingjay are clear, the authors should include baselines in all figures. That’s only the case in Fig. 3 which included results of similar probing models on top of acoustic features, but it would be useful to see how the performance compares against standard supervised approaches at least.

Ablations: report results for encoder+probing models trained from scratch supervisedly instead of pretrain+fine-tune to showcase the effect of self-supervised pretraining.

Selection of tasks: I’m not fully convinced by the results in terms of frame-level accuracy. Why not report WER instead and compare with published results? For the speaker identification case, it is also unclear to me why the authors would perform frame-level speaker identification since speaker identities are global utterance-level factors, i.e. we don’t expect to see speaker-dependent information in individual frames, but rather in their overall statistics. I suggest the authors perform experiments in standard speaker identification/verification settings such as VoxCeleb.

Figure 2: More details are required to parse the figure. Models were pre-trained on the train-clean-360 partitions of LibriSpeech, but which partitions are used for fine-tuning, training, and evaluation?

FIgure 4: Please clarify whether there’s overlap between the pre-training data and the evaluation data (train-clean-360 seems to overlap).

Figure 6: This is a very interesting result. It would be useful to dig a little deeper here to understand why that is the case. Perhaps visualizing attention maps would give a hint as to why the behavior is observed. I intuitively suspect that attention maps become “more local” with depth, and earlier representations are more “long-range-dependent”, which might be beneficial to modeling phonetics.

---

### Official Review · AnonReviewer3 · 2020-06-29
**Re: "Audio ALBERT: A Lite BERT for Self-supervised Learning of Audio Representation"**

**Rating:** 6
**Confidence:** 5

**Review:**

This paper applies ALBERT from NLP in the context of audio, hence the name AALBERT. ALBERT (from NLP) reduces number of parameters by learning a layer and sharing it across multiples layers while maintaining similar performance to BERT. Similarly, AALBERT (ALBERT on Audio) applies similar techniques to Mockingjay (BERT for Audio) to achieve similar down-stream performance with 91% fewer parameters.

Down-stream task evaluation is done by either feature extraction (freezing the pre-trained model and getting a weighted sum of layers as input to the down-stream classifier, as is done in ELMO) or fine-tuning the the whole model.
The tasks that the authors considered are:
1. Phoneme classification task (on LibriSpeech)
2. Utterance-level and frame-level speaker identification tasks

The results of the experiments suggest:
1. AALBERT, having 91% fewer parameters, has comparable performance to Mockingjay.
2. 12-layer AALBERT is performing worst, 6-layer is optimal, while 3-layer is in between. This is unintuitive and the justification of authors is not satisfying.
3. AALBERT and Mockingjay have similar efficiency at using different subset of the training data.
4. t-SNE of the representations show clear separation of the speakers which might indicate the model has learned informative features.
5. Probing experiment: Intermediate layers in AALBERT and Mockingjay are better equipped for these two down-stream tasks than the final layers.

Pros:
- Although the approach, model, or findings are not novel, the execution of the application of the ideas from NLP to audio domain is acceptable.
- Paper is written clearly for the most part.
- Application of ALBERT in audio domain confirms the findings in original NLP domain.
- AALBERT hyper-parameters and training details are sufficient.

Cons:
- Down-stream tasks are limited to only 2 tasks. Although the experiments suggest that the learned representations are capturing useful information, limiting the down-stream tasks to only 2 raises some questions.
- t-SNE of representation for speakers shows separation for only 10 speakers.
- Mockingjay hyper-parameters and training details are non-existence or not as detailed as ALLBERT.

---

### Decision · Program_Chairs · 2020-07-01

**Decision:**

Reject

**Comment:**

Dear author(s),

Thank you very much for your submission at the ICML2020@SaS workshop (https://icml-sas.gitlab.io/). Based on the scores assigned by the reviewers, we regret to inform you that the paper was rejected. We got 26 submissions and we were only able to accept 13 papers. We invite you anyway to consider the feedback of the reviewers and to follow our upcoming workshop on July 17.